# Re-Expression of Poly/Oligo-Sialylated Adhesion Molecules on the Surface of Tumor Cells Disrupts Their Interaction with Immune-Effector Cells and Contributes to Pathophysiological Immune Escape

**DOI:** 10.3390/cancers13205203

**Published:** 2021-10-16

**Authors:** Mostafa Jarahian, Faroogh Marofi, Marwah Suliman Maashi, Mahnaz Ghaebi, Abdolrahman Khezri, Martin R. Berger

**Affiliations:** 1German Cancer Research Center, Toxicology and Chemotherapy Unit Heidelberg, 69120 Heidelberg, Germany; martin.berger.hd@gmail.com; 2Department of Hematology, Faculty of Medicine, Tabriz University of Medical Sciences, Tabriz 5165665931, Iran; marofif@tbzmed.ac.ir; 3Stem Cells and Regenerative Medicine Unit at King Fahad Medical Research Centre, Jeddah 11211, Saudi Arabia; mmaashi@kau.edu.sa; 4Cancer Gene Therapy Research Center (CGRC), Zanjan University of Medical Sciences, Zanjan 4513956184, Iran; sm.ghaebi@sina.zums.ac.ir; 5Department of Biotechnology, Inland Norway University of Applied Sciences, 2418 Hamar, Norway; abdolrahman.khezri@inn.no

**Keywords:** polysialylation, immune escape, electrostatic repulsion, adhesion molecules, apoptosis, metastasis

## Abstract

**Simple Summary:**

The immune system consists of various mechanisms contributing to the battle against cancer cells or hazardous pathogens. However, in cancer progression the immune system is often unable to eliminate neoplastic cells, although immune effector cells infiltrate the tumor environment. The current paper reviews the causes for this immune escape. Specifically, we comprehensively discuss various roles of sialic acids in this process. Specific focus is given to adhesion molecules re-expressed on membranes of tumor cells, which carry oligo- and polysialic acid chains. These carrier proteins loaded with sialic acids direct the interaction between immune effector and tumor cells and thus prevent the “kiss of death” between the latter and the former cells. We also discuss strategies suited to reduce the degree of sialic acid presence on the surface of tumor cells, which can be the basis for future therapeutic intervention.

**Abstract:**

Glycans linked to surface proteins are the most complex biological macromolecules that play an active role in various cellular mechanisms. This diversity is the basis of cell–cell interaction and communication, cell growth, cell migration, as well as co-stimulatory or inhibitory signaling. Our review describes the importance of neuraminic acid and its derivatives as recognition elements, which are located at the outermost positions of carbohydrate chains linked to specific glycoproteins or glycolipids. Tumor cells, especially from solid tumors, mask themselves by re-expression of hypersialylated neural cell adhesion molecule (NCAM), neuropilin-2 (NRP-2), or synaptic cell adhesion molecule 1 (SynCAM 1) in order to protect themselves against the cytotoxic attack of the also highly sialylated immune effector cells. More particularly, we focus on α-2,8-linked polysialic acid chains, which characterize carrier glycoproteins such as NCAM, NRP-2, or SynCam-1. This characteristic property correlates with an aggressive clinical phenotype and endows them with multiple roles in biological processes that underlie all steps of cancer progression, including regulation of cell–cell and/or cell–extracellular matrix interactions, as well as increased proliferation, migration, reduced apoptosis rate of tumor cells, angiogenesis, and metastasis. Specifically, re-expression of poly/oligo-sialylated adhesion molecules on the surface of tumor cells disrupts their interaction with immune-effector cells and contributes to pathophysiological immune escape. Further, sialylated glycoproteins induce immunoregulatory cytokines and growth factors through interactions with sialic acid-binding immunoglobulin-like lectins. We describe the processes, which modulate the interaction between sialylated carrier glycoproteins and their ligands, and illustrate that sialic acids could be targets of novel therapeutic strategies for treatment of cancer and immune diseases.

## 1. Introduction

In cancer progression, the immune system is often not capable of eliminating cancer cells, despite the presence of immune effector cells infiltrating the tumor microenvironment [1,2,3]. In this regard it is important to realize the reciprocal effects between cancer and immune cells, which are modeled to a large extend by carbohydrates present on membrane glycoproteins. Due to their high variability, they orchestrate not only the cellular regulation, but are also involved in processes, such as protein folding, intracellular transport, and immune cell polarization. Furthermore, glycoproteins play a key role in activating the immune system [4,5,6]. However, the poor prognosis of metastatic cancers has been correlated with overexpression of membrane glycoproteins on cancer cells, which are (poly-) sialylated [7,8,9]. In this respect, recent advances in glycobiology and cancer research have described a mechanistic role of sialic acid in tumor development and progression: aberrant sialylation of glycoproteins and glycolipids has been shown to mediate conditions such as increased tumor growth [10,11], inhibition of apoptosis [12,13], metastasis [14,15,16], resistance to therapy [17,18,19], and enhanced invasiveness of tumor cells [11,20]. Specifically, increased sialylation of metastasizing tumor cells [9,21] leads to altered adhesion and changes in transmembrane signaling [22,23]. Therefore, sialic acid has been repeatedly proposed as a possible target against tumor cells [11,20,24,25], ^a)^ of [20] see Appendix A.

## 2. Materials and Methods

### 2.1. Literature Search

This systematic review was carried out according to the Systematic Reviews and Meta-Analyzes (PRISMA) guidelines in order to define the pathophysiological roles of poly/oligo- sialylated adhesion molecules and their co-partners. As indicated in the PRISMA FLOW DIAGRAM, we did an extensive search of the PubMed database (Figure 1). A complete list of the keywords that have been used to search for items are presented in the form of 17 complementary search boxes in Appendix A. The database search was done from 1995 to 2021. The search results were imported into EndNote and the duplicate entries removed based on title and year of publication. (Registration ID 277798).

### 2.2. Data Selection

We have included into the review all items that meet the following criteria:(1)The physiological and pathological roles of membrane adhesion molecules linked with oligo/poly-sialic acid glycosylation in tumor progression, apoptosis, metastasis, angiogenesis, migration, proliferation, and growth of tumors;(2)The biological roles of heterophilic and homophilic membrane adhesion molecules in neuronal and embryonic development, and the development of certain neuronal diseases (such as Alzheimer’s disease, Parkinson’s disease, multiple sclerosis, and schizophrenia) ^b)^.(3)The role of sialic acid in the immune escape of tumors and pathogens (bacteria, viruses);(4)The role of sialic acid in electrostatic repulsion between immune effector and target cells (tumor or pathogens), which re-express membrane adhesion molecules linked with oligo/polysialic acid;(5)The role of sialic acid in differentiation of immune cells (T-cells), and in virus infection;(6)The role of sialic acid receptors/copartners (lectins such as siglecs or other adhesion molecules) in relation to the function of the immune cells;(7)The role of galectins, selectins, and kinases in cooperation with sialylated glycoproteins in or outside of cells.

Entries such as non-English articles, review articles, book chapters, case reports, letters to the editor, and responses were excluded from the study.

### 2.3. Legend to Prisma Flow Diagram

Records were identified by searching the PubMed database for keywords within the period of 1995–2021, which have been listed in the respective Appendix A. The original number of records was reduced by subtracting duplicates, which was followed by selecting those records that focus on the role of sialic acids (especially of poly- and oligo sialic acids) in various biological processes. The final selection was made by concentrating on those records that describe the role of adhesion molecules decorated with poly- and oligo sialic acids that play an important function in embryonal and neuronal development and can be re-expressed on tumor cells for immune escape. 

## 3. Results and Discussion

### 3.1. Structure and Regulation of Sialic (N-Acetylneuraminic)-Acid

Membrane proteins are post-translationally modified by N- or O-glycosylation for improving their molecular stability, for receptor-ligand interactions and ensuing signal transduction [14,26]. Glycoprotein- and ganglioside modifications are differentiated by their terminal glycan structures, e.g., sialic acid- or heparan sulfate- derivatives, which have differential effects for most physiological and pathophysiological functions [27,28,29]. Almost all eukaryotic organisms are able to express sialic acids. This is made possible by sialyltransferases of the glycosyltransferase family 29 (CAZY GT_29), which are widespread in the Deuterostoma lineages and more rarely described in Protostoma, Viridiplantae, and various protist lineages, which have a common ancestor with eukaryotes. In fact, even some pathogenic microorganisms also contain sialic acids, e.g., bacteria, viruses, and parasites; they utilize cell surface sialic acids as ligands to attach to corresponding cell surface lectins, and to infect respective cells [30,31,32,33]. (Some Gram-negative bacteria synthesize sialic acids via an aldolase enzyme: they use a mannose derivative (Man-NAc) as a substrate and attach the three carbons of pyruvate to it, thus resulting in a sialic acid structure [34].) Sialic acid is a key component of glycoconjugates in glycoproteins/glycolipids, which commonly occur in cell membranes, cytoplasm, and glandular secretions that mediate cellular communication [24,35]. The position of sialic acid at the most distal part of non-reducing glycan structures qualifies this α-keto acid to play important roles in biological processes. Sialic acid formation involves more than 20 different sialyltransferases, which transfer sialic acid residues to glycoproteins [36] (Figure 1A,B).

The most common sialic acids are N-acetylneuraminic acid (Neu5Ac), N-glycolylneuraminic acid (Neu5Gc) [37,38], and the deaminated sialic acid derivative 2-keto-3-deoxy-D-glycero-D-galacto-nononic acid (KDN) [13,37,39], in which the fifth carbon atom is substituted by acetoamide/hydroxyacetoamide/hydroxyl groups [40]. Sialic acids (Neu5Ac) may occur in glyco-conjugates, linked as mono sialyl residues via α-2,3 or α-2,6 to galactose or N-acetyl galactosamine at the non-reducing terminal position of glycan chains on glycoproteins and glycolipids, whereas oligo- or polymeric forms of sialylation (oligoSia or polySia) render α-2,8 or α-2,9 glycosidic linkages [5,24,40,41,42]. Their structure and function depend on the linkage type between sialic acid residues and possible substitutions of sulfate/hydroxyl-groups in positions C4/7/8/9 by acetyl/lactoyl/methyl/sulfonyl or phosphonyl groups (Figure 1A,B).

**Figure 1 cancers-13-05203-f001:**
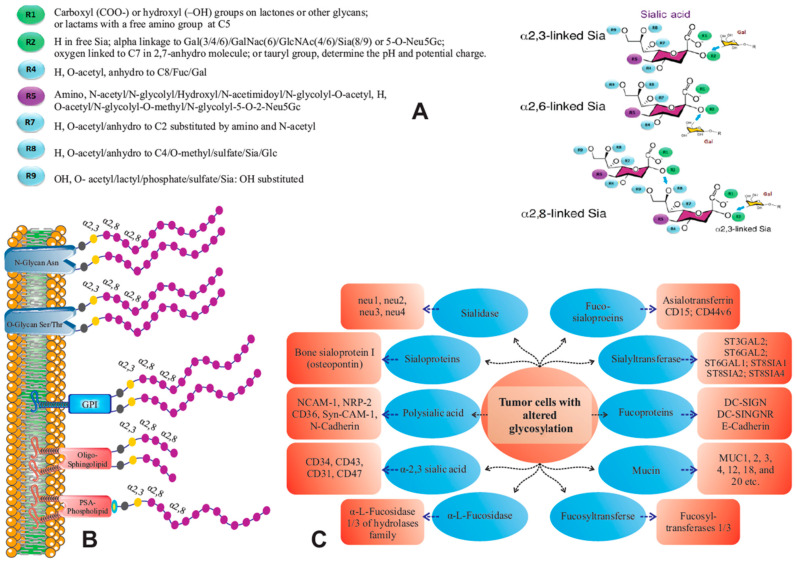
Type of sialic acid linkages and their relation to various carrier molecules. (**A**): The structure of sialic acid (right), which is linked to galactose at positions α2,3 or α2,6, not only protect glycoproteins from uptake and degradation, but also prevent recognition of subterminal Gal and GalNAc residues by asialoglycoprotein receptors [14]. The sialic acid α2,3 linkage can be extended to a sialic acid chain by adding further sialic acid residues to position α2,8. The transfer of sialic acid residues involves more than 20 different sialyltransferases, the most important subgroup being C8 sialyltransferases (ST8Sia I–VI), which transfer a polysialic acid (PSA) chain via an α2,8-linkage to proteins. Sialic acid (Neu5Ac) may occur in glyco-conjugates in α2,3-Sia, α2,6-Sia, α2,8-oligosialylated (3–7), and α2,8-polysialylated (8–200) forms. Various groups, which are attached to sialic acid, are designated as R1–R9 (left). Most common sialic acids are N-acetylneuraminic acid (Neu5Ac)/N-glycolylneuraminic acid (Neu5Gc)/3-deoxy-non-2-ulosonic acid (KDN) [13,37], in which the fifth^.^ carbon atom is substituted by acetoamide/hydroxyacetoamide/hydroxyl groups [40]. Their structure and function depend on the linkage type between sialic acid residues and possible substitutions of sulfate/hydroxyl-groups in positions C4/7/8/9 by acetyl/lactoyl/methyl/sulfonyl or phosphonyl groups. (**B**): Polysialic acid chains (with α2,8 linkages) can be attached to different carriers including O- or N-glycans, GPI anchored proteins, oligosphinglolipids, and phospholipids. The functionally most important polysialic acid chains have a α-helical conformation with dynamic function by their carboxyl groups being directed inwardly and the N-acetyl groups (PolyNeu5Ac and PolyNeu5Gc) outwardly. (**C**): The consequences of altered glycosylations are shown in part (**C**). The scheme indicates that tumor cells with altered glycosylation will show a series of distinctly changed biologic processes.

ST8Sia II and ST8Sia IV sialyltransferases specifically modify carbohydrate chains of certain neuronal cell adhesion molecules, such as NCAM or Neuropilin-2 and SynCam-1 [43,44]. They link up to 200 sialic acid residues by α-2,8-linkage (polysialic acid) and thus endow these molecules a multifunctional role in biological processes including all steps of cancer progression, e.g., cell–cell and/or cell–extracellular matrix interactions, increased proliferation, migration, and decreased apoptosis rate, as well as evasion from immune effector cells and metastasis of tumor cells [45].

The polysialic acid (PSA) chains, which are most flexible, are functionally most important; they have a α-helical conformation with dynamic characteristics caused by their carboxylic group (ionized at physiological pH), which are directed inwards and the N-acetyl groups (PolyNeu5Ac and PolyNeu5Gc) outwards [8,41]. The consequences of altered glycosylations are shown in Figure 1C. 

In contrast to linear macromolecules, like proteins and nucleic acids, carbohydrates can form complex structures. This is based on their physico-chemical reactivity to form branches and their isomeric conformation [46]. When linked covalently to proteins or lipids, the resulting glycoproteins or glycolipids (gangliosides) are endowed with new functions of interaction, e.g., by forming bridges between proteins or proteins and glycolipids (Figure 1B). This complexity corresponds to a code with distinct geometric features that has the ability to transmit information with geometrically progressing proficiency [13,47].

Tumor cells are characterized by a general increase in total sialylation and expression of respective carrier molecules as compared to normal cells [27,48]. These changes in glycosylation have been termed a new hallmark [49,50,51] of malignant transformation. They are driven by altered gene expression, substrate availability, cellular environment, and protein conformation, which can be altered by mutations [27,42]. The degree of sialylation, their type of linkage as well as the length (mono, di, oligo, poly) of the resulting sialic acid chain will determine subsequent effects. The synthesis of PSA is mediated by two dominant polysialyltransferases (ST8Sia II and -IV), which transfer a poly/oligo sialic acid chain via a 2,8-linkage to proteins (resulting to 200 residues with helical form) [13,52].

### 3.2. Physiologic Role of Poly/Oligo-Sialylated Adhesion Molecules and Their Interaction with Growth Factors and Their Receptors

The biological role of sialic acids in relation to the evolutionary perspective of organisms is enormous. Given the breadth of the topic, it is not possible to cover individual topics in detail in this overview, but we briefly describe some areas in which sialic acids play an important role. These include physiological functions as homing of leukocytes via selectins; modulation of the immune system through interaction of sialylated pathogens with siglecs or other receptors located on immune cells; sialic acids as ligands for many microbes or viruses; influence of sialidases or sialic acid transferases on immune cell reactions; modulation of biophysical effects by factor H; suppression of immune cells by apoptosis-inducing sialylated glycoproteins; sialic acid during fertilization (egg-sperm interaction); and sialic acid to control embryonic and neuronal development and learning/cognitive functions [53,54].

The dynamic development of the nervous system depends on processes such as neural plasticity, migration and differentiation of neural precursor cells [55,56,57,58], and the formation of neuronal networks through regulation of genes implicated in neurite growth, guidance, and target recognition. Within this development, the cell numbers are controlled by regulating cell survival and apoptosis, and neural connections are strengthened by the fasciculation of neurites, and the formation, maturation, and plasticity of synapses [56,58,59,60]. All of the aforementioned procedures are regulated to a large extent by a few cell adhesion molecules (CAMs) with specific polySia glycosylation, which possess regulatory properties for homophilic and heterophilic interactions to other cells (trans), or to components of the extracellular matrix (ECM) (*cis* or *trans*) [61,62]. If poly/oligo-sialylated glycoproteins are expressed on both interaction partners, they will inhibit cell–cell and cell–extracellular matrix interactions by steric and repulsive hindrance, caused by their bulky poly/oligo-anionic configuration [63]. They are normally expressed in neural and synaptic cells [e.g., NCAM-1^PSA^, [64] neuropilin-2 (NRP-2^PSA^) [43] SynCAM-1 (CADM-1^PSA^) [65]], in various lymphoid tissues and activated B and T lymphocytes [e.g., C-C chemokine receptor type 7 (CCR7^PSA^ or CD197^PSA^ [43,66]], and in stem cell-derived microglia [e.g., E-selectin ligand-1 (ESL-1^PSA^)] [67]. Other glycoproteins characterized by poly/oligo-sialylation include the α subunit of the voltage-dependent sodium channel [68], the MUC1 protein from the serum of breast cancer patients [69], and CD36 (known as fatty acid translocase or scavenger receptor), which can also be found in human milk [70]. A large number of cells express also polysialyltransferases, which are capable of auto-polysialylation and transferring this PSA-chain to the aforementioned carrier glycoproteins [45,71].

The polysialic acid-associated adhesion molecules are able to interact with their homologous co-receptors and may thus regulate biological processes in the immune defense by homophilic and heterophilic interactions. This is illustrated by, e.g., NCAM-1^PSA^ [55,72] its co-receptor NCAM-2 (RNCAM), SynCAM-1^PSA^ (TSLC1, CADM-1), its respective co-receptors SynCAM-2/3 (CADM2/3 or TSLC2/3) [73], NRP-2^PSA^ (Neuropilin-2 or VEGF165R2) [74,75,76], and its homologous co-receptor NRP-1 [77,78]. Studies on NCAM-2 NCAM-2/OCAM (olfactory cell adhesion molecule, expressed on most cells of the sensory smell system) demonstrate, that the protein is expressed in the human CNS as well as in other tissues [79]. There are three isoforms of NCAM2 located on the membrane, with molecular weights of 90, 115, and 125 kDa, respectively, the smallest isoform of which is attached to glycosylphosphatidylinositol (GPI) on membranes [79,80,81,82,83]. NCAM-2 has a high sequence identity with NCAM-1 [84,85]. The sequence identity between NCAM-1 and NCAM-2 is highest at the Ig1, Ig2, and Ig5 modules and in the cytoplasmic region. This high similarity suggests that the genes encoding NCAM-1 and NCAM-2 are paralogs.

Comparable to NCAM-1 and NCAM-2, the function of polysialylated NRP-2 and its proteoglycan coreceptor NRP-1, which promote binding of the class 3 semaphorins (SEMA3) to their receptor plexinA/B, involves axonal guidance and growth suppressive properties in tumor cells [86,87,88] (Figure 2a). SEMAs were first described as negative mediators of axon pathfinding that repel axons and collapse growth cones [89,90]. Similar to this mechanism, SEMA3F is able to prevent the angio-invasion of tumor cells by repelling epithelial cells [91,92]. This is achieved when SEMA3 forms a complex with both Plexin and NRP-1/-2 [93,94,95]. Besides inhibiting the invasion of blood vessels, the formation of this complex induces a signal, which results in a higher apoptosis rate and further confines cell growth, migration, metastasis, and angiogenesis [91,92,96,97,98]. As SEMA3s and VEGFRs share the same ligands (NRP-1/NRP-2), the binding of SEMA3s constitutes a competitive inhibition of VEGFR binding and thus a negative effect on tumor progression [99,100] (Figure 2a,b).

In this way, NRP1 and 2 interact with SEMA3 membrane-associated receptors located in the central nervous system (plexins) and their growth factors (class 3 semaphorins) [101,102,103]. In addition, they bind to a series of growth factor s and their receptors, including VEGF/VEGFR [75,76,104,105,106], TGF-β1/TGFβ1R [107,108], HGF/c-Met [109,110], PDGF/PDGFR [111,112,113], FGFs/FGFR [112,114,115,116,117], EGF/EGFR, BDNF/TRKA,B [118] as well as IGF/INSR [119]. These interactions favor the growth of potential tumor cells (Figure 3a,b).

Receptor tyrosine kinases (RTK) cannot be activated or induced solely by their respective receptor, as they require additional induction molecules that contribute to receptor activation and even signal transduction. Examples of RTK/co-receptor pairs include EGFR with E-cadherin, alternatively with NCAM-1, [121,122], CDCP1 [123], CD44 [124], L1-CAM (CD171) [125], SynCAM-1 [126], or TSP-1 [127]. Also included is FGFR with Syndecans [128], or, alternatively NCAM-1 [116,129,130,131], TSP-1 [127], as well as VEGFR-2/-3 with NRP-1/NRP-2, or CADM1, NCAM [132,133,134,135,136,137]. Interestingly, VEGFR-2 can also interact with VE-cadherin [138,139], and VEGFR-1 with NRP-1 [133,134,140,141,142,143]. Furthermore, PDGFR can interact with integrins, or NCAM, CADM1 [130,142], CD44 [144], Necl-5 [145], and TSP-1 [127]. Moreover, TGF-β can interact with syndecan-2 [146,147], NRP-1/NRP-2, and NCAM [108]. Further, there is a collaboration between c-Met and several co-receptors, e.g., α6β4 integrin [148], as well as between plexin A/B with NRP-1/NRP2 [43,96] and L-CAM [149]. The interaction of plexin A with NRPs has recently been described in detail [150]. Initially, an interaction is started between a plexin A glycoprotein and a neighboring NRP-1/NRP-2. Subsequently, the NRP–plexin A interaction will prompt the formation of a complex with two NRP-1/-2 and plexin proteins, respectively, which are complemented by binding to a Sema3 factor (Figure 2a). Afterwards, this complex is able to trimerize for intensive signal transduction [102,151]. 

### 3.3. Re-Expression of Polysialylated Adhesion Molecules in Cancer Progression

Besides the physiologic functions of poly/oligosialylated adhesion molecules, they have also pathophysiologic roles. In this regard, quality and quantity and not simply presence or absence play important roles not only in the neuronal and embryonal development, but also in cancer metastasis and some psychiatric diseases. In fact, re-expression of polysialylated adhesion molecules by tumor cells will lead to increased proliferation, plasticity, migration, angiogenesis, metastasis, and downregulation of important adhesion molecules.

From a mechanistical perspective, the negative charges between effector and target cells result from the opposing carboxylate groups of glycoproteins capped with sialic acid. They generate antiadhesive properties [8,56,152,153] by providing an electrostatic repulsive field, which regulates cell–cell interactions in trans, and among adhesion molecules on the cell surface in *cis* [154]. These sialic acids based physiologic mechanisms are also incorporated in pathophysiological traits and support proliferation, migration, and resistance of tumor cells [18,19]. High sialylation levels also protect tumor cells from complement-mediated lysis by preventing antibody binding [155,156] and from phagocytosis by immune cells [156,157]. 

### 3.4. Lectins Are Potential Co-Partners of Sialylated Glycoproteins 

Proteins or glycoproteins with affinity for carbohydrates have been termed lectins, which can be found in plants, animals, viruses, and bacteria [8,158]. Lectins have important roles as intracellular, cell surface, or secreted molecules. Secreted lectins are incorporated to a varying degree into the extracellular matrix, where they regulate biologic processes between cells and their matrix by interactions with external glycans. For example, the viral lectin hemagglutinin is a structural protein of the influenza virus capsid, which can bind to sialic acid residues located on the surface of target cells. For cellular uptake, another structural virus protein, termed neuraminidase, cleaves the glycosidic bond to sialic acid residues, and thus liberates the virus for fusion with the target cell membrane [30,159,160]. In general, lectins can participate in pathological and physiological processes and have different interactions with the immune system, depending on their structure. Typically, lectins contain two or more binding sites for carbohydrate units of proteins, but some may have an oligomeric structure with multiple binding sites [161,162]. The affinity between lectins on the surface of one cell and carbohydrated proteins of neighboring cells is relatively weak, but the sum of interactions is strong due to the resulting sum effect [163].

A key function of lectins in mammals is to serve as adhesion molecules, which facilitate cell–cell interactions. As a result, signal transduction is triggered, including the activation of immune effector cells [164]. Some plant lectins can also serve as potent toxins. The mammalian lectins can be differentiated into classes based on their amino acid sequence and biochemical properties (Table 1). However, lectins in animals serve to interact with existing recognition sites of sialic acids, which initiate a broad range of biological processes and thus affect the complex roles of sialylated glycoproteins [38,165].

One large class are the C-type lectins (C for calcium-dependent). These proteins constitute a superfamily, and its members have in common a domain of 120 amino acids that are responsible for carbohydrate binding. In this structure, a calcium ion links a mannose residue to the lectin and this renders their interaction with co-partners to be calcium-dependent. C-type lectins recognize a vast array of ligands that regulate various physiological functions, including those in the innate and adaptive immune responses. Defects in these molecules lead to developmental and physiological abnormalities, as well as altered susceptibility to infectious and non-infectious diseases [166].

Selectins are members of this family, which are present on epithelial (E), lymphocytic (L), and platelet (P) cells, and by their carbohydrate binding facilitate the homing process of immune-system cells [167,168,169]. All three selectins (heterophilic CAMs) preferentially bind to sulfated and fuco-sialylated derivatives, e.g., PSGL-1 (P-selectin glycoprotein ligand-1), which is a mucin-type glycoprotein expressed on all white blood cells [170,171]. 

### 3.5. I-Type Lectins or Siglecs

Another important class are the I-Type lectins or siglecs (sialic acid-binding Ig-like lectins), which belong to the Ig superfamily. They are characterized by their specificity for sialic acids, which are attached to the terminal regions of cell-surface glycoconjugates. These type 1 transmembrane proteins comprise a sialic acid-binding N-terminal V-set domain, variable numbers of C2-set Ig ligand binding domains, a transmembrane region, and a cytosolic tail (Figure 4). Based on their sequence similarities and evolutionary conservation, two primary subsets of siglecs have been identified: the first subset includes siglec-1 (sialoadhesin), siglec-2, siglec-4 (MAG or myelin-associated glycoprotein), and siglec-15, all of which are well-conserved in mammals [172]. 

Siglec-1 (sialoadhesin), which lacks tyrosine-based motifs, plays a role as a positive regulator of the immune system and is a target for sialylated bacteria, enveloped viruses (e.g., HIV), and other sialylated pathogens [177,178]. 

The second subset of siglecs is designated to the CD33-related siglecs, which potentially inhibit immune response activation by ITIM-dependent signals. ITIM-containing CD33-related siglecs-3 and 5-12 siglecs are negative immunoregulators and endocytotic receptors. A subgroup of these CD33-related siglecs lack the intracellular ITIM but are associated with the adaptor protein DAP-12 (DNAX-activating protein), which contains the immunoreceptor tyrosine-based activating motif (ITAM) [179]. Additionally, CD33 induces apoptosis, and it may also enhance the production of anti-inflammatory cytokines and suppress the production of pro-inflammatory cytokines. According to a recent publication, the overexpression of siglec-9 in macrophages inhibits the production of pro-inflammatory cytokines such as TNFα and enhances the production of the anti-inflammatory cytokine IL-10 in an ITIM-dependent manner in response to toll-like receptor signaling [177,180]. With the exception of siglec-4, in humans, the CD33-related siglecs are expressed differently on various subsets of leucocytes, where they play an important role in the regulation of immune and inflammatory responses as discussed recently [181,182,183]. In this context it is noteworthy that siglec-3, which interacts with polysialylated NCAM-1 (CD56), is a key player in the plaque formation of Alzheimer’s disease [184,185,186,187,188,189,190].

NK and other effector cells express various siglecs, e.g., siglec-3, 7, 8, and 9 [191]. Their presence stabilizes the conformation of membrane glycoproteins in *cis* interactions with endogenous sialoconjugates at the cell surface of NK cells [192]. In addition, they are MHC class I-independent inhibitory receptors on immune effector cells, which—if stimulated—prevent the activation of these cells. However, the expression of corresponding ligands on target cells results in the inhibition of NK cell-mediated cytotoxicity by interacting with siglec-7 and 9 [191]. The outermost position of sialic acids during the glycosylation process implies the capping of galactose and related residues by sialic acids. As a result, these residues are not available for interaction with, e.g., galectins inducing apoptosis [24,193]. 

Siglecs often function as sensor for sialylated glycoproteins [194]. Based on their intracellular ITIM, they induce strong inhibitory signaling when binding to sialic acid [195]. Interestingly, this mechanism is also used by tumor cells and pathogens to escape the immune system, by adding sialic acid residues to their glycan structures, thus highlighting that the sialic acid–siglec interaction is key to the immune function against pathogens and cancer [196,197]. Thus, the, siglecs 14, 15, and 16 play a role in positive and negative immune regulation [161,181,198]. Siglec-14 has an arginine residue in its transmembrane region that is required for its association with DAP12 (ITAM-containing adapter), by which PI3K is recruited [161,181]. It is reported that Siglec-5 and Siglec-14 can be associated for delivering opposing signals via ITIM- and ITAM-dependent pathways, respectively. Another residue of arginine (the first immunoglobulin domain) required for sialic acid recognition by Siglec-5 and Siglec-14 is present in humans, suggesting that these two proteins work cooperatively and balance activating and inhibiting signal transmission through sialic acid recognition [161,181,182].

### 3.6. Sialylation of Check Point Receptors

Studies from anti-inflammatory check point receptors (e.g., CTLA-4 [199,200], PD-1 [200,201], and TIM3 [200,202]) show that an altered degree of sialylation (Figure 1C) influences the interaction with respective co-partners and their proper function [51,203,204,205]. 

The same is true for pro-inflammatory immune checkpoint receptors, such as the TNF receptor superfamily including 1-4-BB, [206] as well as the two stimulatory immune checkpoint proteins CD28 [207] and ICOS (B7-H2) belonging to the B7-CD28 superfamily [205,208,209,210].

It is known that the extracellular domain of ICOS carries three putative N-glycosylation sites [211]. Glycan modification of ICOS is essential for correct folding, trafficking to the cell surface, and ligand binding activity [212]. Other checkpoint molecules like PD-1 and its ligand programmed death ligand 1 (PD-L1, B7-H1) also carry N-glycosylation sites (four in PD-1 and one in PD-L1). Carbohydrate modification of PD-1 is not required for ligand binding, but determines the orientation of PD-1 in vivo and is therefore able to influence its complex formation with PD-L1 [213,214]. PD-L2 is a transmembrane protein expressed in normal tissues to inhibit the activity of T-cells and prevent autoimmunity. PD-L1 is commonly upregulated on the surface of tumor cells, binds to PD-1 expressed on tumor-infiltrating lymphocytes, and eventually causes T-cell tolerance [215,216].

CTLA-4 contains two glycosylation sites in its extracellular domain with low glycan heterogeneity and high production of tetra-antennary N-glycan structures [217]. Low N-glycosylation of CTLA-4 at the cell surface reduces T cell hyperactivity in autoimmunity [197,218].

### 3.7. Sialylated Glycans in Tumor Cells Prevent Galectin Induced Apoptosis, Autophagy, and Cluster Formation

Galectins (S-type lectins) can induce apoptosis, autophagy, and cluster formation of glycoproteins [140,219]. They specifically bind to β-galactosides, such as N-acetyl-lactosamine (Galβ1-3GlcNAc or Galβ1-4GlcNAc), which is linked by N- or O- glycosylation, but this interaction is impeded by increased sialylation of the corresponding glycoproteins present on tumor cells, which reduce their interaction. Other apoptosis-inducing lectins include, e.g., C-type lectins and annexins [162]. Galectins have three different forms, including dimeric (homodimeric galectin 1, 2, 5, 7, 10, 11, 14, 15), tandem (heterodimeric galectin 4, 6, 8, 9, 12), or chimeric structures (galectin-3 exists in monomeric form or, associated via the non-lectin domain, in a pentameric complex). Their stability and ability to bind carbohydrates depends on disulphide bonds. Galectins have a broad variety of functions, including mediation of cell–cell interactions, cell–matrix adhesion, and transmembrane signaling for, e.g., apoptosis regulation [219,220,221]. Galectins participate in controlling the positive and negative selection of T cells in the thymus, as they prevent the circulation of T cells, which would recognize self-antigens by interacting with, e.g., CD43, CD45, or CD7 and then become self-reactive [222,223,224]. Following an immune response to eradicate the excess T cells, epithelial thymic cells secrete both galectin 1 [225,226] and galectin 9, thereby mediating apoptosis in activated or infected T cells [227,228]. Interaction of apoptosis inducing glycoproteins (e.g., CD43, CD 45, CD7, etc.) with galectin 1, galectin 3, or galectin 9 will further initiate different intracellular death pathways (Figure 5a,b) [229,230,231].

The function of many glycoproteins of the cell surface depends on their cluster formation [e.g., integrins [232,233], CD45 [234], TNFR, TRAILR, EGFR [29,235], PECAM [236], and Fas death receptor (CD95) [12], but this function is lost following their high degree sialylation.

The degree of cell surface sialylation is regulated by numerous enzymes, including (i) enzymes that control the synthesis and distribution of the activated sialic acid substrate (i.e., cytidine-5′-monophospho-N-acetylneuraminic *acid* (*CMP*-*sialic* acid)), (ii) the sialyltransferases, and (iii) the sialidases (neuraminidases or neu1-4). In all cancers, altered expression and activity of both sialyltransferases and sialidases is observed [237,238]. These enzymes are typically found in subcellular compartments. The sialyltransferases are located in Golgi and the sialidases in lysosomes, endosomes, and membranes. In all cases, it has been shown that sialylation levels are higher in tumor cells [49,239,240]. Similarly, in macrophages, Fas, and TRAIL receptors, which show increased N-glycan sialylation, they prevent the formation of clusters and thus the binding of apoptosis-inducing ligands [209].

**Figure 5 cancers-13-05203-f005:**
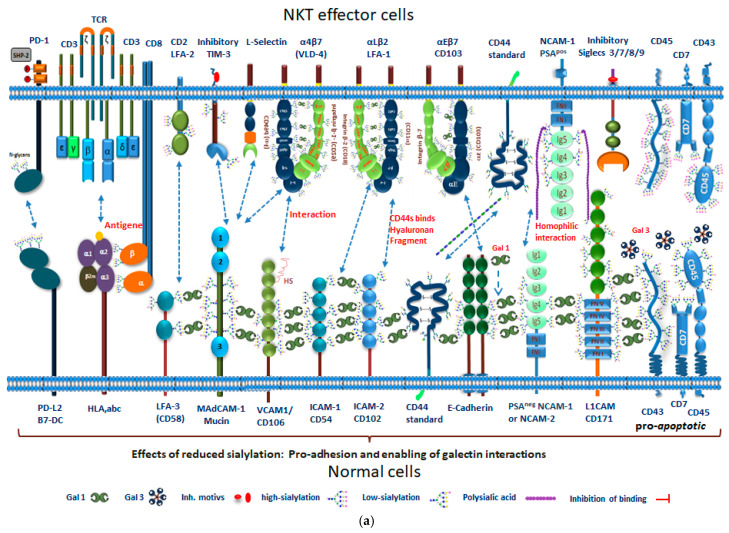
(**a**) Regulatory function of polysialylated glycoproteins in conjunction with adhesion molecules in NKT- effector and non-tumor cells. NKT-effector cells show a high degree of sialylation, which is precondition for their properties, including migration, mobility, and reduced adhesion. The upper part shows examples of glycoproteins present on NKT effector cells carrying either polysialic acid residues (NCAM^PSA^) or α2,3/α2,6 linked sialic acid residues (CD45, CD43, CD7, CD2/LFA2, integrins), inhibitory receptors (siglecs, PD1, TIM3), T-cell receptors (CD3, CD8) as well as sialic acid/heparin sulfate-binding adhesion molecules (CD44, and L-selectin). On normal cells (bottom part), these glycoproteins are opposed by ligands/receptors, which allow interaction with the effector cells, including PD-L1/2, HLA, abc, LFA-3, MADCAM-1, VCAM-1, ICAM-1, CD44s, E-cadherin, NCAM^PSA-neg^, L1-CAM, low sialylated CD7, CD43, and CD45. The normal or down-regulated levels of sialylation will promote target- effector cell interactions, binding of galectins to β-galactosides of glycoproteins, as well as facilitate apoptosis induction by binding of galectin 3 or other ligands to apoptosis inducing receptors (e.g., CD43, CD7, CD45) [241,242]. (**b**) Regulatory function of polysialylated glycoproteins in conjunction with adhesion molecules in tumor and NK-effector cells. NK-effector cells show a high degree of sialylation, which is precondition for their properties, including migration, mobility, and reduced adhesion. The upper part shows examples of glycoproteins present on NK effector cells carrying either polysialic acid residues (NCAM^PSA^) or α2,3/α2,6 linked sialic acid residues (CD45, CD43, CD7, CD2/LFA2, integrins), inhibitory receptors (siglecs, PD1, TIM3, NKGA/B, KIR receptors), as well as sialic acid/heparin sulfate-binding adhesion molecules (CD44 and L-selectin). On tumor cells (bottom part) these glycoproteins are opposed by ligands/receptors, which do not allow interaction with the effector cells, because they are highly sialylated. In addition, many polysialylated adhesion molecules are upregulated (including NCAM, NRP2, SynCAM1, mucins (e.g., MAdCAM-1, CEACAM-1), N-cadherin, L1-CAM, fucosialylated CD44v6) [243]. Inhibitory ligands (PDL1/2, HLA-E/G) show increased affinity for their co-partners on effector cells and thus inhibit the function of the effector cells. High sialylation of CD45, CD7, and CD43 inhibits the binding to galectins and thus protects against apoptosis induction of tumor cells [242].

This will result in the overexpression of the STGal-family members on tumor cells, which are then characterized by increased sialylation (all linkages) and will escape the immune system by blocking multiple signaling pathways leading to apoptosis (e.g., galectin receptors, TNFR1, Fas, TRAIL, etc.) [240,244].

TNFα and FasL, the ligands for TNFR1, TRAILR, and Fas, are mainly expressed by immune cells, which are also a rich source of galectins. The latter are characterized by their binding to receptors (e.g., CD43, CD7, CD29, CD45, and others), which can also induce apoptosis [230,245] (Figure 6a,b). This outcome, however, is prevented in tumor cells as they express sialoproteins or gangliosides, which protect cancer cells from immune effector cells via interaction with inhibitory receptors (e.g., siglecs). 

### 3.8. Polysialylation of Glycoproteins Generates Diverse Functions

Sialic acid plays also an important role in the transport of proteins, amino acids, and ions into cancer cells [38,246,247]. Surface glycoproteins of dendritic cells (DCs) are mostly sialylated. It is suggested that sialylated glycoproteins in DCs have an impact on several aspects of DC biology. Blocking of sialic acid expression in human monocyte-derived DCs (moDCs) by a synthetic, fluorinated sialic acid, which, as a potent mimetic blocks sialic acid expression, enhances the responsiveness of moDCs to toll-like receptor (TLR) stimulation [248]. Additionally, sialidase treatment of DCs improves the efficacy of antigen presentation of DCs, which may be used for vaccines in anti-cancer immunotherapy [66,249,250].

The ensuing sialylation effects include blocking of galectin interactions with galactose residues, which causes changes in protein conformation and stabilization of the spatial distances on cell membranes. This results in altered cellular properties, characterized by anti-adhesion (Figure 5b), cell flow behavior in suspension, metastasis, but also cellular and synaptic plasticity, necessary for cell differentiation and migration, cell growth, and development (e.g., NRP-2/EGFR-complex, Figure 2b) [3,8,18,251,252,253].

Additionally, interactions between sialylated glycoproteins or glycolipids and their ligands (e.g., lectins) regulate important intra- and extracellular signal transduction pathways. These include modulation of the affinity between growth factors and their receptors with consequences for cell differentiation, as well as cell growth and development (Figure 2a,b). Specifically, this interaction alters intra- and extracellular communication related to the initiation of signal transduction for activating the immune system against cancer cells and viral or bacterial pathogens [8]. Mechanistically, the adhesive carrier proteins are located in the area of unsialylated growth factor receptors, which are clustered by galectins and exert an intrinsic capacity to interact with these receptors [251]. 

The additional presence of PSA chains allows for the regulation of signal transduction via influencing the access of ligands to their receptors, thus supporting/regulating the cellular growth and survival as well as angiogenesis [8,117] (Figure 2b). These effects are consequences of functional imbalances between sialic acid residues located on carrier proteins and their lectin or lectin-like binding partners. This ultimately leads to the escape of pathogens (parasitic, bacterial, and viral) and/or tumor cells from the innate/adaptive immune system or even in hyper-reactivity associated with autoimmune and neuronal diseases [196]. Regarding the latter diseases, only a short hint to a vast array of publications seems appropriate in the context of this review.
In fact, numerous publications have described that imbalances in sialic acid distribution and the degree of sialylated glycoproteins in conjunction with their co-partners are causal factors for certain neuronal diseases, such as Alzheimer’s disease [85,188,189,247,254,255], Parkinson’s disease [256], multiple sclerosis [257,258,259], and schizophrenia [260].

It is well known that the absence of sialic acid and galactose residues or the upregulation of certain lectins, e.g., siglec-1 in body cells, leads to various autoimmune diseases and tissue inflammation by activation of immune effector cells [261]. T-cell populations have their own specific surface glycosylation profiles, which are largely responsible for the different susceptibilities of individual subpopulation cells to galectin-1 or galectin-9 -induced cell death [262,263]. The main factor for selection and deletion of T cells during development is galectin-1 induced cell death, which is crucial for the development of distinct subpopulations in the thymus. Low level sialylation of glycans on the surface of CD4/CD8 double positive thymocytes renders this population susceptible, whereas high level sialylation of glycans confers resistance to galectin-1 binding and ensuing apoptosis [225,264]. Low level sialylation of CD43, CD45, and CD7 on T-cell surface allows binding of galectins, and initiates p53/ARF-dependent apoptosis during the process of differentiation [241,265].

Chien et al. published that the differentiation of Th1, Th2, and T-reg cell populations strongly depend on the availability of glucosamine as a substrate for N-glycosylation of CD25, which will impair Th1, Th2 and T-reg populations, but will also enhance Th17 differentiation [203]. The fact that different forms of glycosylation will determine a cell’s fate can be concluded from the following observation: incorrect N-glycosylation of CD25 results in altered IL-2 signaling, which plays an important role in the regulation of T-cell differentiation [203,266,267]. In addition, the core-2 branching of CD45 is associated with increased susceptibility to galectin 1 mediated T cell death [229,268]. However, sialylation of CD45 N-glycans by ST6Gal I inhibits its recognition by galectins, thus preventing the clustering of this receptor tyrosine phosphatase at the cell surface, and rendering the T-cells resistant to galectin-1 mediated cell death [269,270,271]. Various glycosyltransferases of the Golgi apparatus are responsible for posttranslationally modifying the N- and O-glycans, which are crucial for TCRs association with other glycoproteins (MHC-I or II plus antigens) and thus generate the TCR signal transduction and receptor internalization by endocytosis [269]. In summary, TCR, CD28, and CD45 each have multiple N-linked glycosylated sites that interact with endogenous galectin 1 and 3 [225,272].

### 3.9. Proteins with the HNK-1 Epitope Serve a Function Similar to Poly/Oligo-Sialylated Glycoproteins

The human natural killer-1 (HNK-1) epitope is a trisaccharide moiety with a 3′-sulfated glucuronic acid at the non-reducing terminal (HSO_3_–3GlcAβ1–3Galβ1–4GlcNAc-R) and is predominantly expressed in the central nervous system [273]. An important function of this epitope is to facilitate the migration of neural crest cells in the nervous system [274,275]. Proteins involved in the guidance and growth of axons include CAMs [276,277]. The NCAM glycoproteins are important examples of this group. They are a family (e.g., NCAM1, NCAM2) with different splice forms (i.e., different length) and roles. For its function, NCAM can be dimerized in *cis*- or *trans*-positions. In addition, *cis*-dimerized NCAM can form a tetramer by homophilic interaction in trans. For dimerization in *cis*, the Ig1/2 modules interact with each other, for tetramerization, they interact with Ig5 and Ig4, whereas Ig3 interacts with a corresponding Ig3 module [278]. Similar to NCAM1, NCAM-2 may be a marker of certain types of cancer, including human prostate cancer [84,279,280,281]. NCAM-2 is heavily glycosylated on its Ig5 module, which contains four of the eight possible N-glycosylation sites of NCAM-2. Accordingly, glycosylations on NCAM-2 play a role in axon guidance and target recognition [86]. The degree of glycosylation in the Ig5 module probably determines a possible modulation and function of NCAM-2 [282]. Similar to NCAM-2 [283,284], NCAM-1 (except its 180 kDa isoform) carries a HNK-1 carbohydrate structure at the Ig5 module [59,85] (see below).

As HNK1-carbohydrates are also known to modulate cell–cell interactions, the simultaneous presence of more than one carbohydrate epitope may reflect a new mechanism involved in the fine-tuning of NCAM functions. The HNK-1 carbohydrate consists of a small epitope of 3’-sulfated glucuronic acid, which is linked to a lactosaminyl residue that is involved in the homophilic binding of NCAM. [285]. However, the HNK-1 epitope lacks the ability of NCAM^PSA^ to influence the pathways associated with memory consolidation. HNK-1 carbohydrate expression is regulated by the brain-derived neurotrophic factor (BDNF) and the tyrosine kinase receptor (TRKR), which both transmit signals in regenerating motor nerves and promote functional recovery after peripheral nerve repair [286,287].

The extracellular matrix molecule tenascin-R (TNR) also carries the human HNK-1 epitope, which inhibits postsynaptic GABA receptors [288]. Interactions of the HNK-1 epitope with chondroitin sulfate proteoglycans enhance neuronal cell adhesion and neurite outgrowth [287]. The HNK-1 epitope (also referred to as L2 or CD57) [289] is also found in a number of other neural cell adhesion molecules, including L1 [290], close homolog of L1 (CHL1) [291,292] myelin-associated glycoprotein (MAG) [110,290,293], melanoma cell adhesion molecule (MCAM) [293], and contactin [153,294]. 

Remarkably, tumor cells can use proteins with the HNK-1 epitope as support for tumor progression including migration and metastasis. Proteoglycans like NRP-1 or CD44V6 interact with certain integrins (e.g., integrin β1) and tyrosine kinase receptors [295,296,297] modified polyglycosaminoglycan chains (GAG) bearing chondroitin or heparan-sulfate. This modified version of GAG can contain an HNK-1 epitope and is thus able to promote or inhibit cancer cell growth, survival, and invasion [298,299]. 

### 3.10. Polysialylated Glycoproteins Are Co-Receptors for Growth Factors and Their Receptors

An important aspect of α2,8 linked poly/oligosialylated N-glycans of cell adhesion molecules is their ability to form complexes with growth factors and their receptors, thus modulating the respective signal intensity. Besides NCAM-1, the polysialylated carrier glycoproteins include NRP-2 [300,301], and SynCAM-1 [302,303], which both take part in synaptic plasticity of memory consolidation, differentiation, migration, and growth, as well as in cancer progression [300,304,305]. NRP-2^PSA^ is a member of the neuropilin family [75,300], which is expressed on T/DC cells as well as in the central nervous system [306]. NRP-1 and NRP-2 are up to 44% homologous. The sequence homology of human and murine NRP-2 is around 94% [307]. NRP-2 has two major splice variants, which are categorized as NRP-2a and NRP-2b. NRP-2^PSA^ is upregulated in a number of tumor types including osteosarcoma [308], melanoma [309], breast cancer [310], lung cancer [103,311], brain tumors [312,313], colorectal cancer [107,314], pancreatic cancer [315,316,317], myeloid leukemia [132], saliva adenoid cystic carcinoma [300], infantile hemangioma [318], as well as ovary- [100], bladder- [319], and prostate cancers [320] 

The neuropilins are key receptors within the vascular, nervous, and immune systems and can be found on endothelial cells, neuronal axons, and regulatory T cells, respectively. They serve as co-receptors for the plexins in semaphorin binding on neuronal and vascular endothelial cells and for the VEGFRs in VEGF binding on vascular and lymphatic endothelial cells. Hence, they regulate the initiation and coordination of cell signaling by semaphorins and VEGFs [133,321]. 

Syn-CAM-1^PSA^ is another adhesion molecule with an important function in embryonic development [302,303,322], which is re-expressed in T-cell leukemia [323,324], hepatocellular [325], and lung carcinomas [326]. Over-expression of these polysialylated glycoproteins in cancer correlates, as for NCAM-1^PSA^ and NRP-1^PSA^, with an aggressive clinical phenotype. SynCAM-1^PSA^ is expressed on activated NK/CD8^+^ T cells and on dendritic cells. The presence of NRP-2^PSA^, NCAM-1^PSA^, Syn-CAM-1^PSA^ will change the balance of sialic acids on the cell surface and consequently reduce the inter-membrane adhesion between tumor and effector cells. This is achieved through steric hindrance caused by a high density of negative charges that contribute to the hydrated volume of polysialylated glycoproteins [43,327,328].

### 3.11. Polysialylated NCAM-1, NRP-2, and CADM-1 Potentiate Cell Growth Signaling in Tumor Cells and Increase Tumor Progression

The adhesion molecules NCAM-1, NRP-2, and CADM-1 can gain specific surplus functions by the addition of a PSA chain. With the aid of such a PSA chain they can stimulate various growth factor receptors in tumor cells. For example, polysialylated NCAM-1-stimulated migration and proliferation were paralleled by activation of the FGFR and its downstream signaling components. Moreover, NCAM- and FGF-2-mediated FGFR1-signaling in the tumor microenvironment of esophageal cancer stimulated the survival and migration of tumor-associated macrophages and cancer cells [29,309,329].

Epidermal growth factor receptor (EGFR) is a heavily glycosylated transmembrane receptor tyrosine kinase. Upon EGF-binding, EGFR undergoes conformational changes to dimerize, resulting in kinase activation and autophosphorylation and downstream signaling. Increased expression of ST6GalI in cancer cell lines enhances the α2,6-sialylation of epidermal growth factor receptor (EGFR), which increases its tyrosine kinase activity and the phosphorylation of its targets [6,330]. It has been shown that sialylation of epidermal growth factor receptor increases signaling activity and inhibits gefitinib-induced cell death [6].

Cell growth signaling in tumor cells and their progression depend on polysialylated NCAM-1, NRP-2, CADM-1, and some another membrane sialylated tumor proteins (see below as well as Figure 2a,b and Figure 3a,b) [298,331]. Another growth factor receptor, which is modulated in its activity by the presence of PSA chains on tumor associated neuropilins, is VEGFR. A complex formed between VEGFs, VEGFRs, and NRP-1/NRP-2 promotes tumor progression. In line with this, enzymatic removal of PSA from the cell surface led to reduced proliferation and activity of the extracellular signal-regulated kinase (ERK), thus inducing neuronal differentiation of neuroblastoma cells [332].

The respective ligand, human VEGF has five related genes, i.e., VEGF-A, -B, -C, -D, and placental growth factor (PlGF). All isoforms share the same binding sites for receptor tyrosine kinases [333,334]. VEGF-A ligands are bivalent and bind two monomers of their related receptors, i.e., VEGFR1 and/or VEGFR2. The isoforms are capable of binding and cleaving proteoglycans of the extracellular matrix [333] or proteoglycans located on the cell surface (HSPGs) [333]. Heparin and HS increase the affinity of VEGF-forms for VEGFRs and NRPs [134,335]. However, VEGF-A165a forms a ‘bridge’ between VEGFR2 and NRP1/2 by binding both receptors simultaneously [134,336]. Further, VEGF-A and VEGF-C induce the interaction of NRP2 with VEGFR-2 [134,135]. Knockdown of NRP2 expression potently inhibits human endothelial cell migration induced by VEGF-A and VEGF-C [135]. Binding of VEGF to dimerized VEGFR2/3 via Gal-1 or Gal-3 induces angiogenesis [337,338,339]. Tumor cells, which express VEGFRs with α2,6-linked sialic acid residues, are unable to induce angiogenesis, as VEGVRs are capped with sialic acid cannot bind to Gal-1- or Gal-3, which would otherwise favor dimerization of VEGFRs [251,340]. Another mechanism leading to similar results can be observed when polysialylated NCAM or NRP-2 form a complex with VEGFRs via additional factors [341]. This will prevent angiogenesis, but promote growth [135,224], cell signal transduction, and initiation of migration and metastasis via VEGFR-3 [18,134,305,338] (Figure 2b).

Islamov and colleagues have shown that co-transfection of VEGF with GDNF and NCAM, or VEGF with ANG and NCAM, or NCAM plus VEGF, or NCAM plus GDNF into transgenic mice (used as amyotrophic lateral sclerosis model), increased the life-span of the rodents. The results suggest that both approaches enhance the synaptic plasticity (up-regulation of PSD95 and synaptophysin), and support the proliferation, migration, and myelinization of neuron-glial antigen 2 (NG2; chondroitin sulfate proteoglycan 4) positive glia cells [342,343,344]. In contrast, it was shown that abolishment of NCAM in transgenic Rip1Tag2 mice, which are a model for pancreatic β-cell carcinogenesis, induces tumor metastases by upregulating lymph angiogenesis [137]. These authors, however, did not provide data on the sialylation status of NCAM. As has been described, some tumor cell lines express NCAM without polysialylation [345]. 

In summary, the polysialic acid-associated adhesion molecules are involved not only in cell–matrix and cell–cell adhesion and thus control [135,346] growth, migration, metastasis [304,347], vascular permeability [347,348], and block apoptosis in cancer stem cells [17,305,349,350], but also take part also in cell signal transduction [18,134,305]. Additionally, in a complex with growth factor receptors, these molecules control tumor progression [77,304] and regulate angiogenesis.

The important role of Ig-CAMs is revealed by their stimulation of tyrosine kinase receptors for epidermal growth factor (EGF), fibroblast growth factor (FGF), and nerve growth factor (NGF) [351,352]. 

Functions of polysialylated NCAM-1, NRP-2, and CADM-1 in tumor growth and tumor progression influence homo- and heterophilic interactions, which define key intracellular signaling pathways, e.g., by complex formation with FGFR-1, ERK1/2, FAK, and c-Met/ALK [45,353] as has been described recently in neurologic articles. In addition, data from the last fifteen years indicate that tumor cells re-express polysialylated glycoproteins in an analogous manner to bacteria that express polySia [33,354]. Both bacteria and tumor cells profit from the fact that they become resistant to the immune system and therapeutic approaches [14,196,355,356].

The functionality of growth factor receptors is closely linked to the clustering and dimerization of their subunits [118] (Figure 2b and Figure 3a,b). Further, ligand binding leads to receptor dimerization, which induces phosphorylation of the kinase domain and thus its activation. The signaling of these dimers, which respond to respective growth factors, will be amplified by association with oligo- and polysialylated glycoproteins, which are upregulated in tumor cells and function as co-receptors. Different receptors utilize different dimerization/activation strategies. For example, PDGF is a dimer, which cross-links two cell surface PDGF receptor monomers. In contrast, the binding of EGF to its receptor induces a conformational change, which promotes dimerization. Furthermore, FGF is complexed by heparin and crosslinks two FGF monomers. In the case of insulin, the receptor is already dimerized on the cell surface; ligand binding causes a conformational change and autophosphorylation [357].

NCAM specifically binds the glial cell-derived neurotrophic factor (GDNF) via its third immunoglobulin (Ig) domain in the plasma membrane. When NCAM and GDNF receptor GFRα1 are co-expressed, a complex is established via GDNF, which inhibits the homophilic cell adhesion of NCAM [358] and potentiates the role of GDNF in the survival and differentiation of neurons, as well as in malignant neurogenic tumors.

The interaction of brain-derived neurotrophic factor (BDNF) with tropomyosin receptor kinase A/B (TrkA/B) results in the dimerization of the receptor [359,360]. BDNF binds directly to polysialic acid, which caps two N-linked glycans on the Ig5 domain of NCAM and can be transferred to TrkA/B via direct contact or dynamic movement. This complex of BDNF with PSA upregulates growth and/or survival of neuroblastoma cells [118]. 

The transmembrane proto-oncogene tyrosine-protein kinase ROS is encoded by the ROS1 (c-ros oncogene) gene, which originally was discovered as a homologue of the transforming sequence of the avian sarcoma RNA virus UR2 [361]. The structure of ROS1 is similar to the human anaplastic lymphoma kinases ALK and LTK [362,363,364,365]. ROS1 is highly expressed in a variety of tumor cell lines including non–small-cell lung cancer (NSCLC), cholangiocarcinoma, and glioblastoma [361,366]. Abnormal ROS1 kinase activity leads to activated downstream signaling components of several oncogenic pathways including PLCγ, STAT3, PI3K/AKT, VAV3, and MAPK/ERK [367,368,369,370]. When the ROS1 kinase domain is fused to the ligand binding domain of EGFR or TRKA, and is stimulated on the cell membrane by a corresponding growth factor, activation of various combinations of previously noted pathway signaling components [367,371,372] will follow. ROS1 gene fusions (rearrangements) were first identified in a human glioblastoma cell line [368,373,374]. They have been further identified in solid tumors including inflammatory myo-fibroblastic tumor [375,376], cholangiocarcinoma [377], ovarian cancer [378], gastric cancer [379], colorectal cancer [380], angiosarcoma [381], spitzoid melanoma [382], and NSCLC [383,384,385,386]. Other fusion partners include surface proteins such as the novel solute carrier family 34-member 2 gene (SLC34A2), which was described in HCC78 cells [387]. Sequencing of lung cancer tissue revealed the presence of 14 different ROS1 fusion partner genes, including CD74, [366,383,388,389], SLC34A2, [366,383,387], syndecan 4 gene (SDC4) [366,383,388], ezrin gene (EZR) [388,390,391], fused in glioblastoma gene (FIG) [389,392], tropomyosin 3 gene (TPM3) [383,388], leucine-rich repeats and immunoglobulin-like domains 3 gene (LRIG3) [388], gene of KDELR2 (KDEL endoplasmic reticulum protein retention receptor 2 gene) [393], coiled-coil domain containing 6 gene (CCDC6) [394], moesin gene (MSN) [383,386], transmembrane protein 106B gene MEM106B) [365], tumor protein D52 like 1 gene (TPD52L1) [385], clathrin heavy chain gene (CLTC) [384], as well as LIM domain and actin binding 1 gene (LIMA1) [383,395]. Out of this list, CD74-ROS1 occurs most frequently in NSCLC. The break point of ROS1 gene-exons for fusions fusion partner genes are not at the same position [381]. All of the possible breakpoints in ROS1 allow the resulting fusion to preserve the ROS1 kinase domain while also retaining the transmembrane domain of fused ROS1 [388,396] (Figure 3b).

### 3.12. Characterization of NCAM^PSA^


The polysialylated form of NCAM plays a well-known role in brain development and neural plasticity [56,397]. NCAM^PSA^ is an important glycoprotein of the brain, as it makes up around 30% of the weight of the embryonic brain and later still up to 10% of the adult brain [398]. In fact, polysialylated NCAM is present in embryonic phases and during the regeneration of adult neurons. However, it is strongly downregulated in adult neurons [399,400,401]. Interestingly, polysialylation is lost in adult brain tissue, except for areas with cognitive tasks. PolySia expression continues to be detectable in restricted regions only, including the hippocampus, olfactory bulb, amygdala, prefrontal cortex, and hypothalamus, in which neurogenesis and neural modeling continues throughout life [57,60,322]. In addition, NCAM^PSA^ is expressed on monocytes, NK, and NKT cells. 

Re-expression of polysialylated NCAM is found in neural diseases as well as in many neoplasms [402,403]. In the latter type of diseases, presence of NCAM^PSA^ is related to an aggressive clinical phenotype and to metastasis [14,15,404]. For example, NCAM^PSA^ is highly re-expressed in many solid tumors (e.g., colorectal carcinoma, small cell lung cancer [20,405], melanoma [406,407], breast cancer [42], neuroblastoma [408], and other brain tumors such as astrocytoma [409], etc.), as well as in a variety of hematological malignancies including acute myeloid leukemia, acute promyelocytic leukemia [404,410,411], acute lymphoblastic leukemia [412,413,414], NK-cell leukemia, B- and γδ T-cell lymphomas [415,416], etc.

The expression level of polysialylated, oligosialylated, or sialofucosylated adhesion molecules and the quantitative expression of carrier glycoproteins are the main regulators of adhesion and polarization between tumor and immune effector cells [9,40]. They inhibit cell–cell and cell–matrix interactions [305,417]. The polysialic acid-associated adhesion molecules mask the cell surface by their repulsive effect and downregulate other adhesion molecules, for example, ICAM and E-cadherin [121,345,418,419]. E-Cadherin and ICAM inhibit the migration of cells. Basically, both proteins are adhesion molecules, but E-cadherin is part of cell–cell junctions, and its presence therefore inhibits the separation of a cell from the environment. In slight contrast, ICAM binds to integrins and is partly responsible for the interaction between effector and target cells. A downregulation of E-Cadherin and ICAM through re-expression of NCAM-PSA or other PSA carrier adhesion molecules would therefore be expected to facilitate migration and metastasis of tumor cells.

For cell separation from muscle cluster tissue, downregulation of E/N-cadherin and upregulation of PSA are sufficient, thus reducing adhesion [420]. Further, they block the cytotoxic interaction of effector cells, and by siglecs can even induce apoptosis in neighboring effector cells. These combined effects are responsible for the immune escape of tumor cells expressing polysialylated adhesion molecules. 

Tumor cells benefit from α2,8 poly/oligosialylated carrier proteins as their surface is masked and their apoptosis signaling is thus blocked (Figure 6b) [337,421]. Further, this results in a physical anti-adhesion effect between the effector and target cells. Moreover, the tumor cells may influence the effector cells by inhibiting their cytotoxic function via inhibitory ligands (siglecs or checkpoint proteins; Figure 4 and Figure 5b) or by induction of pro-apoptotic molecules (e.g., tumors upregulate CD70, which is a ligand for the apoptosis inducing protein CD27 on lymphocytes) [183,422]. In addition, in tumor and effector cells, the presence of α2,8-linked poly/oligosialic acid residues prevents homophilic interaction of adhesion molecules, as well as receptor ligand binding, e.g., for FAS and TNFR with FAS-L, galectins, and TNFα, which otherwise would induce apoptosis in these cells. However, their presence can elicit apoptosis in neighboring cells via trans-homophilic and heterophilic interactions [209,423] (Figure 6b).

The respective tumor cells seem to gain stem-cell-like properties including increased migratory and metastatic potential as well as a higher drug resistance to chemotherapy treatment [17,19]. They are masked towards effector cells (immune escape) and become resistant to anticancer treatment as the uptake of negatively charged drugs will be reduced [424]. In addition, hyper-sialylation of respective adhesion molecules inhibits binding to specific galectins (Gal 3 and Gal 9), which would induce apoptosis in the tumor cells [9,421]. In addition to this external mechanism, an intracellular mechanism is based on, e.g., sialylation of the Fas receptor, which prevents the initiation of the death receptor complex [220,423].

### 3.13. NCAM-1^PSA^, NRP-2^PSA^ or SynCAM-1^PSA^ Downregulate the Expression of Adhesion Molecules

A major obstacle to an endogenous anti-tumor immune response is poor infiltration of TILs into the tumor mass. The migration of immune cells into tumors can be hindered by many factors, such as an impaired chemokine expression in the tumor environment, the reduction of adhesion molecules in tumor cells, tumor- related endothelial cells, or leukocytes, and can be responsible for a defective monitoring of the immune system.

Cytotoxicity studies involving co-incubation of polyclonal natural killer (NK) or clonal *NK*-*92* (NK tumor cell line) effector cells with NCAM-deficient or NCAM-transfected tumor cells showed strong reduction in NK-mediated lysis of tumor cells overexpressing NCAM^PSA^ [345]. This can be explained by NCAM^PSA^ overexpression-induced reduction of adhesion molecule expression, e.g., ICAM or E-cadherin [23,345,425]. This reduction, in turn, will inhibit the interaction and polarization of target and effector cells [345].

An effective mechanism of the tumor associated endothelial cells is the down regulation of adhesion molecules such as ICAM-1/2, VCAM-1, E-selectin, P-selectin, and MAdCAM-1 to prevent immune cell trafficking into the tumor site [426,427,428,429] (Figure 5a,b).

Moreover, the interaction of the PSA chains with inhibitory siglecs will block the effector cell activity or may induce apoptosis in the respective immune cells. In contrast, tumor cells being NCAM-deficient or expressing NCAM without polysialylation and cytoplasmic domain showed high susceptibility to NK lysis [345]. This effect was corroborated by an experiment in which the interaction of NCAM^PSA^ tumor cells with NK cells was blocked by an NCAM-specific monoclonal antibody, which abolished the anti-adhesive properties of the PSA chains and thus normalized the NK cell mediated lysis of the target cells expressing NCAM^PSA^ [345]. These two mechanisms highlight the key role of NCAM in the immune escape of tumor cells expressing NCAM^PSA^. 

NCAM^PSA^ interacts in heterophilic form in *cis* or trans with a series of surface glycoproteins including chondroitin sulfate, heparan sulfate proteoglycans [430], L1-CAM [431,432], E/N-cadherins [419,433], integrins [13,22], DC-SIGN [434], growth factor receptors [131], and their ligands. Heterophilic interaction of NCAM^PSA^ with the glycoproteins noted above can modulate their interaction intensity with corresponding proteins from opposing cells, without affecting their intrinsic binding properties [55,433,435]. 

Furthermore, the interaction of PSA chains located on carrier glycoproteins with the noted glycoproteins can change their function, which then contributes to the progression of many cancer types [153,436,437]. Heterophilic and homophilic interactions of NCAM at the cell membrane in *cis* and trans depend on the presence of PSA. In comparison with CD2/LFA-3 and LFA-1/ICAM-1 interactions, which are involved in NK cell cytotoxicity [438], NCAM^PSA^ on NK cells can negatively influence NK cell cytotoxicity against tumor cells expressing carrier glycoproteins decorated with PSA [345]. Proof of this fact can be found in the observation that antibodies against LFA1, LFA3, and ICAM-1 block tumor cell lysis caused by NK cell mediated cytotoxicity [438,439]. Anti-NCAM mAbs added to peripheral blood mononuclear cells (PBMC) inhibit cytokine-induced killer cells (CIK), which indicates an essential role of NCAM molecules expressed on NK cells in either alloantigen recognition or delivery of accessory signals to CD8^+^ T cell precursors [440].

An example for this modulation has been described for DC-SIGN (CD209), which interacts with weakly polysialylated NCAM in *cis* or trans. DC-SIGN on macrophages interacts with mannose type carbohydrates, being part of pathogen-associated molecular patterns commonly found on viruses, bacteria, and fungi [441]. This interaction induces phagocytosis of pathogens. DC-SIGN is a C-type lectin and has high affinity for ICAM3 (CD50) and DC-SIGNR (CD299) [442], but a low affinity to weakly polysialylated NCAM-1 [434]. DC-SIGN may bind various microorganisms by recognizing high-mannose-containing glycoproteins on their envelopes. It further functions as a receptor for several viruses such as HIV, Lassa, Ebola, and Hepatitis C [443,444,445]. Binding to DC-SIGN can promote HIV and hepatitis C viruses to infect T-cells via dendritic cells [444,446]. Lysis of HIV infected dendritic cells by polysialylated NCAM^dim^ NK cells will be increased by the presence of an anti-DC-SIGN antibody, which inhibits the interaction between DC-SIGN and its ligand, and thus favors the interaction between DC-SIGN and NCAM-1 [434]. This in turn will decrease the repulsion from adhesion molecules in trans (neuropilin 2 on the surface of dendritic cells and NCAM on the surface of NK cells) and thus increase the cytotoxicity of these NK-cells [447]. Expression of NRP-2 is up-regulated in dendritic cells during maturation, coincident with increased expression of ST8Sia IV [447]. Additionally, activated effector cells of the adaptive immune system express concomitantly or alternatively polysialylated NRP-2, SynCAM-1 (CADM1), or polysialylated NCAM-1 [448]. 

Therefore, the presence of polysialylated glycoproteins on the surface of target and effector cells prevents interaction between dendritic cells and T lymphocytes. Removing PSA from NRP-2 or blocking NRP-2 by specific antibodies promotes interaction between dendritic and T cells [447]. Similar to this procedure, using an anti-NCAM^PSA^ antibody modulates NK-mediated lysis of target cells expressing polysialylated NCAM [345].

Another example is given by the *cis* interaction of NCAM^PSA^ with E-cadherin, which reduces the homophilic E-cadherin-mediated cell adhesion in trans [418,419] and thus promotes increased cell migration of pancreatic cancer cells. Cadherins are transmembrane proteins that mediate cell–cell adhesion, depending on the site of expression [419,449,450]. E-cadherin is instrumental in controlling cell polarity and organization of epithelial cells during embryonic development [436,449,451,452]. N-cadherin is normally down regulated during development and is absent in regenerated adult nerve fibers [453].

Cancer cells, which re-express polysialylated adhesion molecules like NCAM-1, NRP2, or SynCAM-1 by oncogenic K-ras will not only downregulate E-cadherin but also prevent homophilic interaction of E-cadherin-mediated cell adhesion in trans and thus promote metastasis [349,361,418,454,455]. Enzymatic removal of PSA from NCAM^PSA^ or blocking polysialylation leads to increased E-cadherin-mediated cell–cell aggregation and decreased cell migration [418,419].

Mechanisms contributing to immune evasion are summarized below. As stated above, adhesion molecules including ICAM and E-cadherin are downregulated in tumor cells in response to the re-expression of NCAM1, NRP3, or SynCAM1, and therefore the polarization between tumor infiltrating lymphocytes (TILs) and solid tumor cells is interrupted [3,355]. TILs contain chemoattracted infiltrating immune effector cells expressing high levels of granzyme B and other highly sialylated markers, for instance mucin-like leukosialin/CD43, CD34, or high sialylated lamp-family such as Macrosialin (CD68) and/or polysialylated adhesions molecules such as NCAM (CD56), NPR-2, and SynCAM-1. These immune effector cells include CD8(+) T cells, CD56(+) NK cells, CD56(+) NKT cells, and CD68+ macrophages [3,355,456,457]. The consequence of interactions, resulting from effector cells (expressing NCAM-1^PSA^) and tumor cells (re-expressing NCAM-1^PSA^) is an anti-adhesion effect caused by electrostatic repulsion.

### 3.14. Dual Role of NCAM-1^PSA^


NCAM-1^PSA^ has a key role in neuronal synapse development, but its re-expression in tumor cells promotes progression. NCAM^PSA^ contains a heparin-binding domain, which after binding to heparan sulfate is vital for its developmental role in synapses, correspondingly, the binding of heparin to this site inhibits NCAM-1 polysialylation and thus prevents migration, invasion, and transition to resistant tumor cells [247,458,459]. 

Enzymatic removal of PSA or of heparan sulfates from neuronal cells diminished the formation of synapses on respective neurons, suggesting that interaction of NCAM-1^PSA^ with heparan sulfate proteoglycans, e.g., CD44, is necessary for their function. In line with this, transfection of NCAM-1^PSA^-deficient neurons with an NCAM-1 isoform carrying PSA stimulated synapses formation on NCAM-1 isoform-expressing neurons [460].

L1-CAM (CD171) belongs to the immunoglobulin superfamily and is widely present on the cell surface of post-mitotic neurons, on axons of post-migratory neurons, and on glial cells [461]. Physiologically, L1-CAM is moderately expressed on the surface of immune cells, at moderate to low density on normal cells, and at intermediate density on many human tumor cells, including melanoma and neuroblastoma, carcinomas from lung, kidney, and skin, and monocytic leukemias [408,461,462,463]. In vertebrates, the family of L1-CAM (cell adhesion molecules) includes four structurally related transmembrane proteins: L1, close homolog of L1 (CHL1), NrCAM, and neurofascin. L1-CAM is a sialic acid-binding lectin and together with NCAM plays a role as signaling coreceptors in neuronal migration and process outgrowth [461,464]. L1 is an enhancer of integrin-mediated cell migration [465]. NCAM-180 and L1-CAMs are co-receptors of integrin- and GDNF receptor signal transduction [466]. Extracellular domains of L1-CAM have six Ig-like C2-type domains, five fibronectin type-3 domains with 21 potential N-linked glycosylation sites. L1-CAM can have homophilic interaction or heterophilic association with neurocan, phosphocan, laminin, integrins [αVβ3, αIIβ3, and α5β1] [467,468], CD9 [469], CD24 [464,470], NCAM-1 [461], neuropilins [454,471,472], FGFRs [473], and proteoglycans containing chondroitin sulphate, etc. [431,455]. 

Cell adhesion molecule L1 disrupts E-cadherin-containing adherens junctions and increases scattering and motility of MCF7 breast cancer cells [474,475]. The cytoplasmic domain of L1-CAM interacts intracellularly with ankyrin and kinases [476]. L1-CAM plays a role in kidney morphogenesis, lymph node architecture, T cell co-stimulation, neurohistogenesis, and homotypic interaction, and thus plays an important role in nervous system development, including neuronal migration, cell to cell adhesion, and differentiation [408,435,477]. NCAM-1-transfected cells, depending on polysialylation, alter their L1-CAM expression. This aspect hints to the directive role of NCAM-1 towards L1-CAM expression [345,478]. The complex formation between polysialylated NCAM-1 variants and L1-CAM in *cis* is the basis of synaptic plasticity, memory formation, and damage repair in the central nervous system (CNS) [55,461,479]. An NCAM^PSA^/L1-CAM or NRP-2 ^PSA^/L1-CAM interaction may significantly influence the role of L1-CAM regarding the tumor cells’ adhesion or migration [480]. In summary, regulation of L1-CAM, E-cadherin, and ICAM-1 by re-expressed NCAM-1 ^PSA^, NRP-2 ^PSA^, or Syn-CAM ^PSA^ plays a crucial role in the immune escape mechanism of advanced carcinomas.

### 3.15. Perspectives and Therapeutic Potential

Conventional cancer therapies such as surgery, chemotherapy, and radiotherapy failed to cure most types of cancer, especially when the disease has reached a metastatic state. Therefore, new cancer treatment methods are desperately needed. One of these could be immunotherapy, which promotes the activation of the immune system against demasked tumor cells [481,482].

The pathophysiological mechanisms outlined above suggest a treatment strategy that will interfere with tumor cell properties resulting from hyper-sialylation. This concept can be realized by reducing the density of sialic acid residues present on tumor cells. Reduced expression of PSA on cancer cells will result in the demasking of cell surface glycans characterized by sulfated oligosaccharides (e.g., the HNK-1 epitope) [483], which can act as tumor suppressors [484] and thus support a therapeutic strategy that is based on a sensitization of cancers to the immune system [248].

For this purpose, multifunctional antibodies can be used, which on the one hand mark re-expressed oligo- or polysialylated carrier proteins on tumor cells [356] and link them to infiltrating effector cells or oncolytic viruses and also show sialidase activity for removal of sialic acid residues, or carry a lectin-like peptide for neutralizing the anti-adhesive properties of sialic acid residues [8]. On the other hand, nanoparticles/vesicles are equipped with a tumor targeting device and are loaded with factors, which knock down components of the process leading to oligo- or poly-sialylation. Finally, viral vectors infecting tumor cells can be used, which cause re-expression of sialidases or knockdown of oligo/poly-sialic acid transferases [248].

In our opinion, the most effective method against the immune escape of tumor cells is based on reducing the link between carrier molecules and sialic acids. This may be achieved by blocking sialylation via either directly suppressing the activity of sialic acid transferases, or with competitive inhibition by presenting an alternative substrate competing for the same binding site. This treatment does not only interfere with a single protein or pathway, but targets a mechanism, which is common to metastasizing tumor cells and immune escape.

## 4. Conclusions

Sialic acids (neuraminic acids), which are located at outermost positions of carbohydrate chains linked to specific glycoproteins or glycolipids play a central role in immune regulation. In addition, changes in protein sialylation of cell surface are one of the most important characteristics of tumor cells. The most important form is formation of poly-sialic acid chains (α-2,8-linkages), which are attached to carbohydrate scaffolds of cell adhesion molecules by ST8Sia II and ST8Sia IV sialyltransferases. This characteristic property correlates with an aggressive clinical phenotype and endows them with multiple roles in biological processes that underlie all steps of cancer progression, including regulation of cell-cell and/or cell-extracellular matrix interactions, as well as increased proliferation, migration, reduced apoptosis rate of tumor cells, angiogenesis and metastasis. 

Furthermore, upregulation of poly- and oligosialylated carrier proteins triggers down-regulation of important adhesion molecules as e.g., of ICAM-1 and E-cadherin. The respective tumor cells gain stem cell like properties including increased migratory and metastatic potential as well as drug resistance. They are masked towards effector cells (immune escape) and become resistant to anticancer treatment because of reduced drug uptake as well as loss of apoptosis induction.

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
