# Peer review of "Re-Expression of Poly/Oligo-Sialylated Adhesion Molecules on the Surface of Tumor Cells Disrupts Their Interaction with Immune-Effector Cells and Contributes to Pathophysiological Immune Escape"

_cancers, 2021, doi:10.3390/cancers13205203_

Round 1

Reviewer 1 Report

The article is substantially unmodified with respect to the original submission.

Since my criticisms concern the pertinence of a book chapter to the journal Cancers, I give the Editor the responsability of the final decision.

Author Response

The article is substantially unmodified with respect to the original submission.

Since my criticisms concern the pertinence of a book chapter to the journal Cancers, I give the Editor the responsibility of the final decision.

We regret that the referee has the opinion that our manuscript is substantially unmodified with respect to the original submission. We had inserted changes into our article, which aimed at the original argument of the referee, by reducing the content and by updating the coverage of the literature.

We do not reject the opinion of the referee that our article could also serve as book chapter to the journal Cancers. Here, we rely on the editor’s judgement, whether our article should be published as book chapter or as review.

Reviewer 2 Report

The article by Jarahian et al provides a detailed and well-written update on the role of glycans, in particular sialic acid, in mechanisms that drive cell interactions. The authors should be commended for the extent to which they have integrated the various aspects of this complex process in their manuscript.

I have a few minor points to raise, which I hope will be useful to improve the manuscript before publication.

A strong focus of the article is on the interaction between tumor and immune cells, but this is not apparent at all in the title of the manuscript. Should the title better reflect this focus?

Please correct a typo in the Author list, where the number referring to the affiliation of author “Abdolrahman Khezrie” is mistakenly indicated as “e”.

In the Flow diagram, authors should describe more precisely how the selection of 1% articles was performed to be considered as a representative sample (eg random selection or other more targeted approach…). This could be done as part of a legend to the diagram or within the Materials and Methods text.

I could not find any mention of the flow diagram in the main body of the text. It probably would be best to mention it at the start of the Materials and Methods section.

In the Abstract (line 34), I would suggest modifying the start of the newly incorporated paragraph like so: “More particularly, we focus on polysialic acid chains,…”, rather than “Here, we focus on…”

Lines 35-40, I would suggest a few modifications to make this new section easier to read. This sentence is very long and should be split into two shorter ones. The first sentence could conclude with a full stop after SynCam-1 (line 37). The second sentence could also be slightly modified for clarity: “It endows them with multiple roles in biological processes that underlie all steps of cancer progression, including cell-cell and/or cell-extracellular matrix interactions, increased proliferation, migration…”

Line 113, please start the sentence with : “The most common sialic acids….”

Line 124-128. This sentence is largely a copy/paste of the sentence in lines 35-40 and should be slightly rephrased to avoid exact repeat.

Lines190-193, sequence identity and homology are not perfectly interchangeable terms. The authors should make sure that homology is only used to imply common ancestry between genes/proteins. For example, line 190, saying “NCAM-2 has a high sequence identity with NCAM-1 [84,85]”, rather than “a high sequence homology”, would be more appropriate. Similarity may be used in line 191 to avoid repeating identity. Also, lines 192-193, do the authors want to say that the high sequence identity mentioned above suggests these genes are paralogs/homologous?  

Lines 309-310: I do not completely understand the following sentence: “The outermost position of sialic acids during the glycosylation process implies the capping of a variety of glycosylation structures”. Does this mean to say that sialic acids themselves cap other glycosylation structures? If so, this may be said a bit more explicitly.

Lines 657: “E-Cadherin and ICAM inhibit the migration of cells.” This is a blunt statement that could be a bit more refined by explaining the role of these proteins in cell/cell junctions (eg E-cadherin), which as a consequence means that they act as negative modulators of migration.

 Line 658: “A downregulation of E-Cadherin and ICAM through re-expression of NCAM-PSA or other PSA carrier adhesion molecule certainly facilitates migration and metastasis of tumor cells.” Do the authors refer to specific published results (in which case the corresponding studies should be cited) or to the anticipated consequences of Ecad or ICAM downregulation? If the latter is true, the sentence could be tweaked a little to read: “ A downregulation of E-Cadherin and ICAM through re-expression of NCAM-PSA or other PSA carrier adhesion molecule would therefore be expected to facilitate migration and metastasis of tumor cells.”

Line 759, maybe this paragraph should start like so: “Mechanisms contributing to immune evasion are summarised below.”

Author Response

Referee 2

 We thank the reviewer for his helpful critique and detail our changes as follows.

  1. A strong focus of the article is on the interaction between tumor and immune cells, but this is not apparent at all in the title of the manuscript. Should the title better reflect this focus?
    We have changed the title as recommended by the referee and accordingly have altered the abstract to reflect this change.

  2. Please correct a typo in the Author list, where the number referring to the affiliation of author “Abdolrahman Khezrie” is mistakenly indicated as “e”.
    We have corrected the typo to read as ‘5’ now.

  3. In the Flow diagram, authors should describe more precisely how the selection of 1% articles was performed to be considered as a representative sample (eg random selection or other more targeted approach…). This could be done as part of a legend to the diagram or within the Materials and Methods text.
    We thank the referee for this suggestion and have followed it by adding a description of our selection process into Materials and Methods and by adding a legend to the Flow Diagram.

  4. I could not find any mention of the flow diagram in the main body of the text. It probably would be best to mention it at the start of the Materials and Methods section.

    This was done

  5. In the Abstract (line 34), I would suggest modifying the start of the newly incorporated paragraph like so: “More particularly, we focus on polysialic acid chains,…”, rather than “Here, we focus on…”

    This was done

  6. Lines 35-40, I would suggest a few modifications to make this new section easier to read. This sentence is very long and should be split into two shorter ones. The first sentence could conclude with a full stop after SynCam-1 (line 37). The second sentence could also be slightly modified for clarity: “It endows them with multiple roles in biological processes that underlie all steps of cancer progression, including cell-cell and/or cell-extracellular matrix interactions, increased proliferation, migration…”

    We thank the reviewer for this suggestion and have altered these sentences accordingly. The new paragraph reads as follows:
    ‘ Tumor cells, especially from solid tumors, mask themselves by re-expression of hypersialylated neural cell adhesion molecule (NCAM), neuropilin-2 (NRP-2), or synaptic cell adhesion molecule 1 (SynCAM 1) in order to protect themselves against the cytotoxic attack of the also highly sialylated immune effector cells. More particularly, we focus on α-2,8-linked polysialic acid chains, which characterize carrier glycoproteins such as NCAM, NRP-2 or SynCam-1. This characteristic property correlates with an aggressive clinical phenotype and endows them with multiple roles in biological processes that underlie all steps of cancer progression, including regulation of cell-cell and / or cell-extracellular matrix interactions, as well as increased proliferation, migration, reduced apoptosis rate of tumor cells, angiogenesis and metastasis.’

  7. 113, please start the sentence with : “The most common sialic acids….”

    This was done.

  8. Line 124-128. This sentence is largely a copy/paste of the sentence in lines 35-40 and should be slightly rephrased to avoid exact repeat.

    We thank the reviewer for this comment. We have addressed this issue by changing the wording in lines 35-40, as indicated above, and hope that this helps avoiding exact repeat.

  9. Lines190-193, sequence identity and homology are not perfectly interchangeable terms. The authors should make sure that homology is only used to imply common ancestry between genes/proteins. For example, line 190, saying “NCAM-2 has a high sequence identity with NCAM-1 [84,85]”, rather than “a high sequence homology”, would be more appropriate. Similarity may be used in line 191 to avoid repeating identity. Also, lines 192-193, do the authors want to say that the high sequence identity mentioned above suggests these genes are paralogs/homologous?  

    We thank the reviewer for this clarification. We have altered the text accordingly and yes, we want to state that the two genes are paralogs.

  10. Lines 309-310: I do not completely understand the following sentence: “The outermost position of sialic acids during the glycosylation process implies the capping of a variety of glycosylation structures”. Does this mean to say that sialic acids themselves cap other glycosylation structures? If so, this may be said a bit more explicitly.

    We thank the reviewer for this question. In response, we have modified this sentence to read now as follows:
    ‘The outermost position of sialic acids during the glycosylation process implies the capping of galactose and related residues by sialic acids. As a result, these residues are not available for interaction with e.g. galectins inducing apoptosis.‘

  11. Lines 657: “E-Cadherin and ICAM inhibit the migration of cells.” This is a blunt statement that could be a bit more refined by explaining the role of these proteins in cell/cell junctions (eg E-cadherin), which as a consequence means that they act as negative modulators of migration.

    We are thankful for this suggestion and have altered our wording as follows:
    ‘ Basically, both proteins are adhesion molecules, but E-cadherin is part of cell / cell junctions and its presence inhibits therefore the separation of a cell from the environment. In slight contrast, ICAM binds to integrins and is partly responsible for the interaction between effector and target cells. ‘

  12. Line 658: “A downregulation of E-Cadherin and ICAM through re-expression of NCAM-PSA or other PSA carrier adhesion molecule certainly facilitates migration and metastasis of tumor cells.” Do the authors refer to specific published results (in which case the corresponding studies should be cited) or to the anticipated consequences of Ecad or ICAM downregulation? If the latter is true, the sentence could be tweaked a little to read: “ A downregulation of E-Cadherin and ICAM through re-expression of NCAM-PSA or other PSA carrier adhesion molecule would therefore be expected to facilitate migration and metastasis of tumor cells.”

    We thank the reviewer for this suggestion and have altered our wording accordingly.

  13. Line 759, maybe this paragraph should start like so: “Mechanisms contributing to immune evasion are summarised below.”

We thank the reviewer for this suggestion and have altered our wording accordingly.

This manuscript is a resubmission of an earlier submission. The following is a list of the peer review reports and author responses from that submission.

Round 1

Reviewer 1 Report

This is a well written review over a broad subject and includes a lot of different receptors and molecules that interact with sialylation. 

Comments:

  • In the simple summary it is nicely explained that the focus is on poly/ oligo sialylated adhesion molecules and partners and what this is. However, in the abstract there is no mention of poly or oligo sialylation, what it means. There is only mentioning about sialylation in general. I would recommend to include this in the abstract, as the whole review is based on the poly/oligo sialylation.
  • The materials and methods clearly explain which data is included and a nice figure is representing this in a flow diagram. In the diagram it does not become clear why records are excluded, or why only 2450 records from 625544 articles were screened. 
  • In section 2b the section that starts at line 211. This is really messy and not clearly written and gives detailed information that is not necessary for this part and makes it harder to read.
    • For example: line 220; the sequence identity between NCAM-1 and NCAM-2 is highest at.... 
    • The next paragraph starts with on the other hand. I am still wondering why because there is no opposite part.
    • Figure 2A shows no clear interaction between the components, while this is clear in 2B
    • In Figure 2a/B the explanation of the different symbols is missing. While this is nicely added to 3A.
  • Table 1: misses MGL as a protein. Furthermore, aberrant expression of MGL is described in glioblastoma cancers 
  • On line 295 there is stated that C-type lectin is one large class, but then only a few lines follow and not really into dept. While the other classes that are smaller are described in a large part. Please make this more comparable and in balans. 
  • At line 312, all of a sudden a new paragraph starts, this should be one part with the last paragraph. 
  • At line 325, In humans this subset includes. Which subset is this?
  • At the end of section 2e the conclusion is drawn that Siglec-14, 15 and 16 play a role in positive and negative immune regulation. There is no foundation for this mentioned in the whole section to only name Siglec-14 till 16 and not for example Siglec-7 or Siglec-9.
  • Line 391: sentence starting with "Similarly in...", this sentence is not correct English.
  • In section 2K all of a sudden it switches from NCAM and CADM1 to ROS production on line 604. It is clear that it is involved in cancer. However, I don't think it should be this elaborately discussed in this review as nothing is mentioned in combination with sialylation. 
  • In line 695 has a couple of typo's with numbers within the words.
  • Line 752, it here states that adaptive therapy show tumor infiltrating lymphocytes and that they play an important role in adaptive immune system. This is kind of redundant to say here, as T and B cells are part of the adaptive immune system. It would be better to delete this sentence and conclude with the last sentence in this section. 
  • Section 2N from line 770 is confusing because here human and vertebrate are mentioned both and one sentence is about vertebrate and the other human. Please make this more clear as it is confusing as the human part is called L1-CAM and the vertebrate part is called L1. 
  • Figure 1C has a typo in sialic acid
  • Figure 4: A lot of the inhibitory Siglecs are also present on DCs, not only Siglec-1.

Author Response

Reviewer 1

  1. In the simple summary it is nicely explained that the focus is on poly/ oligo sialylated adhesion molecules and partners and what this is. However, in the abstract there is no mention of poly or oligo sialylation, what it means. There is only mentioning about sialylation in general. I would recommend to include this in the abstract, as the whole review is based on the poly/oligo sialylation.

    We have added a paragraph to the abstract, in which we describe the role of PSA.
    This reads as follows: ‘We focus in this review on the role of oligo- and polysialic acid chains linked to certain adhesion molecules, which following re-expression in tumor cells cause a malfunction of the immune system by modulating all steps of cancer progression, particularly with respect to the apoptosis rate, cell-cell and/or cell-extracellular matrix interactions including evasion from immune effector cells, and increased proliferation, migration and metastasis of tumor cells.’

  1. The materials and methods clearly explain which data is included and a nice figure is representing this in a flow diagram. In the diagram it does not become clear why records are excluded, or why only 2450 records from 625544 articles were screened. 

    The reduction from 625544 to 2459 records was accomplished by selecting only about 1% of the total articles as a representative sample (n = 6300 articles). This number was further reduced by selecting articles describing adhesion molecules decorated with oligo / polysialylated acid chains. This intermediate step is now added to the figure.

  2. In section 2b the section that starts at line 211. This is really messy and not clearly written and gives detailed information that is not necessary for this part and makes it harder to read.
    For example: line 220; the sequence identity between NCAM-1 and NCAM-2 is highest at.... 
    The next paragraph starts with on the other hand. I am still wondering why because there is no opposite part.
    Figure 2A shows no clear interaction between the components, while this is clear in 2B
    In Figure 2a/B the explanation of the different symbols is missing.
    While this is nicely added to 3A.

    We have altered the respective parts as requested by the referee.
    a: The text was altered to read now: 2 is highest at the Ig1 (line 191)
    b: This was altered to read now: Comparable to NCAM-1 and NCAM-2, the function (line 194)
    c: The Figure was improved, it shows now the interaction between the components.
    d: The explanation of symbols was added to Figure 2A/B

  1. Table 1: misses MGL as a protein. Furthermore, aberrant expression of MGL is described in glioblastoma cancers
    We have added MGL to Table 1, as suggested by the referee.

  1. On line 295 there is stated that C-type lectin is one large class, but then only a few lines follow and not really into dept. While the other classes that are smaller are described in a large part. Please make this more comparable and in balans. 

We thank the referee for his advice regarding our description of the C-type lectins. We have followed his advice and added information on the role of these molecules to keep the description in balance with other families of the lectins. The new parts are highlighted in yellow. The part reads as follows:

‘One large class are the C type lectins (C for calcium-dependent). These proteins constitute a superfamily and its members have in common a domain of 120 amino acids that is responsible for carbohydrate binding. In this structure, a calcium ion links a mannose residue to the lectin and this renders their interaction with co-partners to be calcium-dependent. C-type lectins recognize a vast array of ligands which regulate various physiological functions, including those in the innate and adaptive immune responses. Defects in these molecules lead to developmental and physiological abnormalities, as well as altered susceptibility to infectious and non-infectious diseases 201.
Selectins are members of this family, which are present on epithelial (E), lymphocytic (L), and platelet (P) cells, and by their carbohydrate binding facilitate the homing process of immune-system cells 201-205. All three selectins (heterophilic CAMs) preferentially bind to sulfated and fuco-sialylated derivatives e.g., PSGL-1 (P-selectin glycoprotein ligand-1), which is a mucin-type glycoprotein expressed on all white blood cells 206-209.

  1. At line 312, all of a sudden a new paragraph starts, this should be one part with the last paragraph. 
    We thank the referee for this advice and in response we have connected the two parts.

  2. At line 325, In humans this subset includes. Which subset is this?
    We thank the referee for the advice and have changed the sentence as follows:

‘With the exception of siglec-4, in humans, the CD33-related siglecs are expressed differently on various subsets of leucocytes, where they play an important role in the regulation of immune and inflammatory responses as discussed recently.’

  1. At the end of section 2e the conclusion is drawn that Siglec-14, 15 and 16 play a role in positive and negative immune regulation. There is no foundation for this mentioned in the whole section to only name Siglec-14 till 16 and not for example Siglec-7 or Siglec-9.

    We thank the referee for this advice and in response we have added the following information:
    ‘Siglec-14 has an arginine residue in its transmembrane region that is required for its association with DAP12 (ITAM-containing adapter), by which PI3K is recruited 195, 215. It is reported that Siglec-5 and Siglec-14 can be associated for delivering opposing signals via ITIM- and ITAM-dependent pathways, respectively. Another residue of arginine (the first immunoglobulin domain) required for sialic acid recognition by Siglec-5 and Siglec-14 is present in humans, suggesting that these two proteins work cooperatively and balance activating and inhibiting signal transmission through sialic acid recognition195, 215.’

  2. Line 391: sentence starting with "Similarly in...", this sentence is not correct English

We have corrected the wording as follows:

‘Similarly, in macrophages, Fas and TRAIL receptors, which show increased N-glycan sialylation, prevent the formation of clusters and thus the binding of apoptosis-inducing ligands 16.’

  1. In section 2K all of a sudden it switches from NCAM and CADM1 to ROS production on line 604. It is clear that it is involved in cancer. However, I don't think it should be this elaborately discussed in this review as nothing is mentioned in combination with sialylation. 

    We have addressed this point as follows:
    ‘Epidermal growth factor receptor (EGFR) is a heavily glycosylated transmembrane receptor tyrosine kinase. Upon EGF-binding, EGFR undergoes conformational changes to dimerize, resulting in kinase activation and autophosphorylation and downstream signaling. Increased expression of ST6GalI in cancer cell lines enhances the α2,6-sialylation of epidermal growth factor receptor (EGFR), which increases its tyrosine kinase activity and the phosphorylation of its targets 6, 386. It has been shown that sialylation of epidermal growth factor receptor increases signaling activity and inhibits gefitinib-induced cell death.6
    Cell growth signaling in tumor cells and their progression depend on polysialylated NCAM-1, NRP-2, CADM-1 and some another membrane sialylated tumor proteins (see below as well as Figures 2a,b and 3a,b) 167, 354, 387-389. Another growth factor receptor, which is modulated in its activity by the presence of PSA chains on tumor associated neuropilins is VEGFR. A complex formed between VEGFs, VEGFRs and NRP-1/NRP-2 promotes tumor progression. In line with this, enzymatic removal of PSA from the cell surface led to reduced proliferation and activity of the extracellular signal-regulated kinase (ERK), thus inducing neuronal differentiation of neuroblastoma cells 390.

  2. In line 695 has a couple of typo's with numbers within the words.
    The typos in line 703 (formerly 695) have been corrected

  3. Line 752, it here states that adaptive therapy show tumor infiltrating lymphocytes and that they play an important role in adaptive immune system. This is kind of redundant to say here, as T and B cells are part of the adaptive immune system. It would be better to delete this sentence and conclude with the last sentence in this section.
    We thank the referee for this advice and in response we have altered our wording as follows:
    ‚In summary, the following mechanisms contribute to the immune evasion: As stated above, adhesion molecules including ICAM and E-cadherin are downregulated in tumor cells in response to the re-expression of NCAM1, NRP3, or SynCAM1, and therefore the polarization between tumor infiltrating lymphocytes (TILs) and solid tumor cells is interrupted3, 417. TILs contain chemoattracted infiltrating immune effector cells expressing high levels of granzyme B and other highly sialylated markers, for instance mucin-like leukosialin/CD43, CD34 or high sialylated lamp-family such as Macrosialin (CD68) and / or polysialylated adhesions molecules such as NCAM (CD56), NPR-2 and SynCAM-1. These immune effector cells include CD8(+) T cells, CD56(+) NK cells, CD56(+) NKT cells and CD68+ macrophages 3, 417, 523, 524. The consequence of interactions, resulting from effector cells (expressing NCAM-1PSA) and tumor cells (re-expressing NCAM-1PSA) is an anti-adhesion effect caused by electrostatic repulsion. ‘

  4. Section 2N from line 770 is confusing because here human and vertebrate are mentioned both and one sentence is about vertebrate and the other human. Please make this more clear as it is confusing as the human part is called L1-CAM and the vertebrate part is called L1.

    We thank the referee for this advice and in response we have changed our wording to L1CAM in each case, i.e. also for vertebates.

  5. Figure 1C has a typo in sialic acid
    We thank the referee for this hint. In response, we have corrected the typos in Figure 1C

  6. Figure 4: A lot of the inhibitory Siglecs are also present on DCs, not only Siglec-1.
    We thank the referee for this advice and in response we have complemented the Figure 4

Reviewer 2 Report

The paper is a very unusual review article because it looks like a book chapter. The authors declare that they started from a literature search spanning 30 years and concerning an extremely wide topic. With such approach, they placed over 500 references, sometime connecting articles distant very many years and concerning several biochemical aspects poorly or not related to cancer. To my opinion, in the present form the article is far from the aims of the Journal and the interests of its readers.

I suggest to rewrite the article focusing on the literature of the last 10-15 years and concerning a well defined and specific topic, of direct relevance to cancer, not previously focused by recent review articles, to be discussed in a more critical and specific manner. General biochemical aspects as well relation to other diseases should be only mentioned, quickly referring to recent review articles concerning such topics.

Author Response

Reviewer 2:

The paper is a very unusual review article because it looks like a book chapter. The authors declare that they started from a literature search spanning 30 years and concerning an extremely wide topic. With such approach, they placed over 500 references, sometime connecting articles distant very many years and concerning several biochemical aspects poorly or not related to cancer. To my opinion, in the present form the article is far from the aims of the Journal and the interests of its readers.

I suggest to rewrite the article focusing on the literature of the last 10-15 years and concerning a well defined and specific topic, of direct relevance to cancer, not previously focused by recent review articles, to be discussed in a more critical and specific manner. General biochemical aspects as well relation to other diseases should be only mentioned, quickly referring to recent review articles concerning such topics.

We thank the reviewer for his comments and agree that we submitted an unusual review article. In order to explain our approach, we wanted to review the multifunctional role of sialic acid across several disciplines, and highlight this role in cancer progression. Our finding is that there are similar functions, which have implications in different biological processes including the embryonal and neuronal development.

We agree that a book chapter seems very appropriate, because we are authoring a complementary article focussing on the sialic acid linkages, which have not been included here. This does, nevertheless, not contradict a publication of this review in the current journal.

The special focus of our article on cancer progression suggests, at least in our eyes, that the selected journal ‘Cancers’ is highly suited for this publication and the readers will profit from our description that the effects of sialic acid in general and polysialic acid especially (i.e. electrostatic repulsion and downregulation of adhesion molecules), have bearing on the interaction of immune effector and tumor target cells, including the aspects of apoptosis, proliferation and growth, as well as migration and metastasis.

With regard to the time period, which we cover by respective literature, we preferred in many instances to select the original articles for citation, which are older and then span a longer period than suggested by the reviewer.

To address this concern, we have reduced the number of ‘old’ articles and replaced these by newer ones. In addition, we have omitted two topics of our article, covering lactoferrin (2o) and upregulation of sialic acid under hypoxic conditions (2p). By this we have reduced our literature by 107 references. This allowed also, to span a lower time period of the referenced articles.

With these changes we have met the criticism of the reviewer as much as possible for us. We hope that this will help to reach a positive decision.